# Infection Due to *Actinobacillus lignieresii* in Cattle with Brain and Ocular Involvement: Histological and Microbiological Features

**DOI:** 10.3390/vetsci10050311

**Published:** 2023-04-24

**Authors:** Antonio Salvaggio, Renato Paolo Giunta, Maurizio Percipalle, Fabrizio Scalzo, Anna Maria Fausta Marino

**Affiliations:** Experimental Zooprophylactic Institute of Sicily, 90129 Palermo, Italy

**Keywords:** *Actinobacillus*, granuloma, cattle, brain, ocular globe

## Abstract

**Simple Summary:**

*Actinobacillus lignieresii* was identified for the first time by Lignieres and Spitz in 1902, who listed it as the causative agent in a case of bovine granulomatosis in Argentina. The bacterium is a well-known normal habitant of the oral cavity of ruminants and causes opportunistic infections. The primary organs involved are the mouth, tongue and pharynx. In this study, we describe a case of cerebral and ocular metastatic diffusion of granuloma due to infection with *Actinobacillus lignieresii* in cattle.

**Abstract:**

The actinobacillosis is rare and to date the biological profile of the agent is not yet fully understood. The knowledge about the possible hosts of the pathogen is incomplete and is generally only associated with granulomatous lesions in cattle and sheep. The primary organs involved are the mouth, tongue and pharynx. Human infection is extremely rare. *Actinobacillus lignieresii* is the causative agent of a rare bovine granulomatous disease known as “wooden tongue”. In this research, we describe a case of cerebral and ocular metastatic diffusion of granuloma due to infection with *Actinobacillus lignieresii* in cattle, probably resulting from primary oral localization. Diagnosis was made using histopathological assay which made it possible to highlight the typical lesion of actinobacillosis, and bacteriological assay that allowed to isolate the pathogen.

## 1. Introduction

*Actinobacillus lignieresii* is an aerobic, non-motile, nonspore-forming, Gram-negative coccobacilli that is widespread in soil and manure and is found as normal flora in the respiratory and upper gastrointestinal tract of ruminants [1]. *A. lignieresii* was identified for the first time by Lignieres and Spitz in 1902, who listed it as the causative agent in a case of bovine granulomatosis in Argentina [2].

The bacterium is a well-known normal habitant of the oral cavity of ruminants and causes opportunistic infection. The primary organs involved are the mouth, tongue and pharynx. Contrary to the Actinomycetes, which attack skeletal tissues [3], *Actinobacillus* has a tendency to invade soft tissues, such as lungs, abdominal viscera and regional lymph nodes [4].

Human infection is extremely rare and the infection occurs through ingestion of infected bovine milk or after a direct transfer of the pathogen through broken skin [5]. In the literature, there are only a few cases of human actinobacillosis [6].

We describe a case of cerebral and ocular metastatic diffusion of granuloma due to infection with *Actinobacillus lignieresii* in cattle. This unusual localization of *A. lignieresii* in cattle resulted in epidemiological conditions similar to the other forms of the disease. It has also been confirmed that the route of entry of the pathogen can be decisive in its clinical manifestation.

## 2. Case Presentation

A dairy Holstein Friesiancow in its late stage of life, aged about 12 years, was slaughtered for suspected BSE after the onset of neurological symptoms, such as staggering, locomotor ataxia, depressed sensorium and unilateral exophthalmos. The examination of the carcass showed the presence of an oval neoformation of about 5 cm in diameter within the oral cavity (Figure 1). The oral lesion was not detected during clinical examination because it was located deep in the upper jaw bone at the *sulcus palatinus*; in fact the oral lesion was not ulcerated.

After cutting the skull along a paramedian sagittal axis, a similar neoformation below the protruding eyeball, also involving the optical nerve, was observed; both neoformations seemed to consist of a fibrous tissue of medium consistency, with pinkish and punctuate yellowish-white areas (granulomas) scattered throughout its thickness. This aspect also affected the brain tissue in its entirety (Figure 2). A careful examination of all the other organs and systems did not reveal additional macroscopically appreciable lesions. The test for diagnosis of BSE performed on the brainstem, according to the procedure established by the current European legislation, rendered negative results.

A specimen of cerebral granulomatous tissue and both neoformations was fixed in 10% neutral-buffered formalin, embedded in paraffin, processed routinely and stained with hematoxylin and eosin, periodic acid-Schiff (PAS) and Gram stain. Histological examinations were performed according to our standard laboratory procedures and processed using Tissue Processing Center TPC 15 Duo (MEDITE^®^) embedded in paraffin wax (Medite tissue wax 56–58 °C). Five μm thick histological sections were cut using microtome (Reichert Jung 1150 Autocut) and collected on glass slides (Menzel Gläser, Thermo Scientific, Waltham, MA, USA). The sections were stained with hematoxylin and eosin, periodic acid-Schiff (PAS), Gram stain and observed under optical microscope (Leica DM750) to identify potential morphological alterations. Photographs were produced using an optical microscope (Leica DMLB, Monument, CO, USA) equipped with a digital camera (Leica DFC500, Monument, CO, USA).

Histological examination revealed a loose connective tissue with scattered foci of granulomatosis. The granuloma is characterized by a mass of small aggregates of bacteria surrounded by numerous neutrophils, macrophages, plasma cells and some giant cells and by a wall of granulation tissue which developed into a connective tissue. The tissue was surrounded by eosinophilic club-like bodies. In addition to the granulomatous reaction, the Splendore–Hoeppli phenomenon was observed, characterized by hyaline, acidophil (Figure 3), and PAS-positive material (Figure 4) which is arranged as a crown of clavate protrusions around each colony. Gram stain was then carried out and results were Gram-negative bacteria.

The bacterial culture was performed by operating an initial seeding in blood agar incubated at 37 °C for 48 h, starting with the swabs obtained from different granulomatous areas. The suspected colonies of primary culture, which were sticky, non-hemolytic, small, raised, smooth and grayish-white, were isolated in blood agar. *Actinobacillus lignieresii* is a pleomorphic Gram-negative coccobacillus with bipolar staining. The colonies which resulted to be oxidase positive and catalase positive were then seeded in agar, where a complete profile of acidification was observed. The microorganism was able to grow on MacConkey agar and was found to be urease positive, indol negative and Voges–Proskauer negative. In addition, it fermented glucose, xylose, sucrose, maltose, mannitol and arabinose, reduced nitrate and was beta-galactosidase positive. Trehalose and lactose were not fermented.

Most commercially available identification systems do not contain *A. lignieresii* in their database, so the final identification of the pathogen was performed using Biolog Microstation ID System (Biolog, Hayward, CA, USA). This is one of the automated systems with a database containing information for the identification of several *Actinobacillus* species, and it is a frequently used tool in both human and veterinary microbiology laboratories. The Biolog System tests the ability of bacteria to utilize 95 diffeent carbon sources [7]. The identification of the test bacterium can generally be achieved within 24–48 h by comparing its utilization pattern (metabolic fingerprint) to the Biolog database [7]. Bacterial suspensions were prepared by removing bacterial colonies from the plate surface with a sterile swab and agitated in fluid C, which indicated the presence of fastidious bacteria. The bacterial suspension was adjusted to 68% of turbidity and was the dispensed into Biolog GN III plate. The plates were incubated for 24 to 48 h at 37 °C and then read with the Biolog Microstation 3.5 software. This software compared the results obtained with the tested strain to the database and provided species-level identification based on distance calculation.

For an accurate diagnosis, it was necessary to supplement the histopathological examination, which has shown the typical lesion of actinobacillosis, with the bacteriological examination which allowed to isolate the pathogen.

## 3. Discussion

*A. lignieresii* has been isolated in a variety of species. It is ubiquitous and causes sporadic disease, primarily in cattle, sheep, and goats. Since the organism is a commensal of the oral cavity of ruminants, development of disease is most frequently associated with damage to the oral mucosa. However, sheep, that use their lips to take food, frequently have lesions in lips and cheeks.

Actinobacillosis or pyogranulomatous glossitis induces skin lesions in the parotid region and throat at the level of the tongue which enlarges and hardens, and that is whu it is known as wooden tongue. *A. lignieresii*, usually found in soil and a normal inhabitant of the mouth of cattle, can enter the skin or mucous membranes through cuts or abrasions caused by rough feed or traumatic injury. Usually the cases are sporadic, but in some farms the disease is more frequent [8].

The most frequent clinical signs include inflammation of the gums; hard and swollen tongue, which may protrude from the mouth causing food to fall out due to the inability of the tongue to move properly; dehydration and weight. Granulomas may also form on other parts of the body due to the spread of the infection to the skin. Enlarged lymph nodes in the head and neck region are often evident (atypical or cutaneous actinobacillosis) [8]. The location of the lesions actually suggests that the primary lesion is probably located in the lymph nodes and subsequently extends to the subcutaneous tissue and skin, causing ulceration. Hence, this form of actinobacillosis referred to as atypical or cutaneous [8], should actually be referred to as lymphatic actinobacillosis as the primary lesion appears to occur in the lymphatic system. The limited research present in the literature suggest that lymphatic actinobacillosis is widespread in the southern regions of South America [4]. In some of these studies, the disease was associated with the feeding of dry fibrous forage which could injure the mucous membranes of the oral cavity, favoring the colonization of *A. lignieresii* [9]. In other studies performed on beef cattle, instead, the disease occurred in a herd that grazed in green pastures [10]. Therefore, the frequency of actinobacillosis lesions in the lymph nodes of the head and neck was similar in cattle from different production systems, suggesting that feeding with fibrous foods represents a risk factor for lymphatic actinobacillosis, however, other factors should be considered, according to Lignierès and Spitz [2]. These authors suggested that the high frequency of the disease may have been associated with previous outbreaks of the foot-and-mouth disease, probably because the virus-induced oral lesions favored the bacterial infection, or as a result of the stress and immunosuppression caused by the viral disease. In another study, actinobacillosis was associated with the presence of mud on the cows’ udders as a consequence of high environmental humidity [11]. The same authors have also found outbreaks in young cattle, suggesting incomplete permanent dentition as a risk factor [11], but other authors reported outbreaks in adult cattle with complete dentition, indicating that the disease occurs in cattle of different ages [4,11]. In South America, where lymphatic actinobacillosis is particularly common and affects the parotid, submandibular and retropharyngeal lymph nodes, this bacterium has been shown to be a major cause of granulomatous lesions [12]. In fact, the authors demonstrated that many of the lesions macroscopically diagnosed as tuberculosis are actually actinobacillosis or actinomycosis. These data highlight the importance of the differential diagnosis between actinobacillosis and tuberculosis, and therefore the importance of carrying out histological and microbiological examinations of macroscopic lesions for pathological and etiological confirmation [12]. Numerous studies suggest that lymphatic actinobacillosis is much more common than other forms of the disease and may occur at a higher incidence than other forms of actinobacillosis. Furthermore, the clinical signs and the absence of concomitant lesions in the oral mucosa in most of the affected animals suggest that the bacterium penetrates through the intact mucosa and spreads through the lymphatic vessels, localizing mainly in the lymph nodes of the head and neck [4,12].

The definition of cutaneous actinobacillosis should instead be used in cases where the lesion is associated with infection of skin wounds by *A. lignieresii* which can occur in different anatomical sites. In cattle, in particular, they can occur in the head and neck and in the limbs which is rather rare. It is hypothesized that the presence of lesions on the limbs may be due to transmission through infected saliva during allogrooming or autogrooming, and that the pathogen enters the skin through small abrasions or percutaneous wounds. This type of transmission is frequent in young cattle [13].

The localization of *A. lignieresii* in the gastric or respiratory tract, on the other hand, is related to the spread of the bacterium through the oropharyngeal or nasopharyngeal route together with food or mucus. To date, all authors certainly agree on the importance of isolating and treating affected animals as soon as possible after the initial diagnosis, since contamination of the environment by infected animals can contribute to the transmission of actinobacilosis [13].

Some authors have drawn attention to the importance of molecular diagnosis for the identification of *A. lignieresii*, especially in atypical cases of the disease [14]. Scheid et al. [14] found that actinobacillosis with hippopotamus-like face clinical manifestation, although unusual, occurs with epidemiological conditions similar to the other forms of the disease and that the route of entry of the agent into the infection may be determinant in its clinical manifestation.

In this study, we showed a cerebral and ocular localization of infection by *Actinobacillus lignieresii* in cattle, probably resulting from primary oral localization. Strains of *A. lignieresii* are reported to vary in their ability to cause the disease, both in natural and experimental infections [13], so the peculiarity of this case lies in the fact that *A. lignieresii* has shown a particular tropism for the CNS as opposed to the soft tissue surrounding the injection site, an event never described until now, and which might make one suspect the emergence of a neurotropic strain of the bacterium, with a significant increase in its pathogenicity. The zoonotic factor stands out as the most dangerous when one considers that of the very few human cases reported, two of the three fatal ones manifested meningeal involvement, showing how for humans the CNS tropism of *A. lignieresii* in some ways has already been demonstrated. The involvement of the nervous tissue opens new scenarios both for the zoonotic aspect of the disease and for intra vitam differential diagnosis with the classic nervous system diseases, such as encephalitis and other encephalopathies. The diagnosis was certainly supported by the positivity to the Splendore–Hoeppli phenomenon, which highlighted an intensely eosinophilic, club-shaped material that radiated around the bacteria. The material is usually composed of antigen–antibody complexes, debris and fibrin. The phenomenon is observable in infections of different nature (fungi, bacteria and parasites) including actinobacillosis [15]. It has been shown that the deposition of immune complexes, and therefore of Splendore–Hoeppli positive material, causes tissue damage primarily by activating the complement cascade and activating neutrophils and macrophages via their Fc receptors. Activated leukocytes release prostaglandins, chemotactic substances, oxygen free radicals and lysosomal enzymes which include proteases capable of digesting collagen [16].

## 4. Conclusions

This research has made it possible to highlight an unusual localization of *A. lignieresii* in cattle, with clinical manifestation of cerebral and ocular involvement and it is confirmed that the route of entry of the agent may be decisive in its clinical manifestation.

Concerning the treatment of infected animals, it is important to begin treatment early, as it has been noted that in animals in which the infection is very advanced treatment may not have an effect. However, to prevent relapses, already treated animals should be observed regularly.

Regarding the prevention of infection with *A. lignieresii*, there are no vaccines available to treat this disease. Control is achieved by early recognition and prompt treatment or cases, and isolation or disposal of infected animals is recommended.

## Figures and Tables

**Figure 1 vetsci-10-00311-f001:**
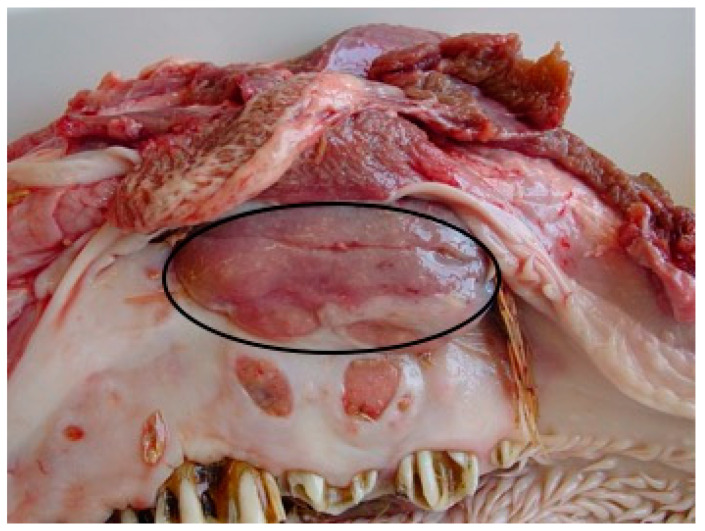
Neoformation within the oral cavity (circle). The oral lesion was located deep in the upper jaw bone at the *sulcus palatinus*.

**Figure 2 vetsci-10-00311-f002:**
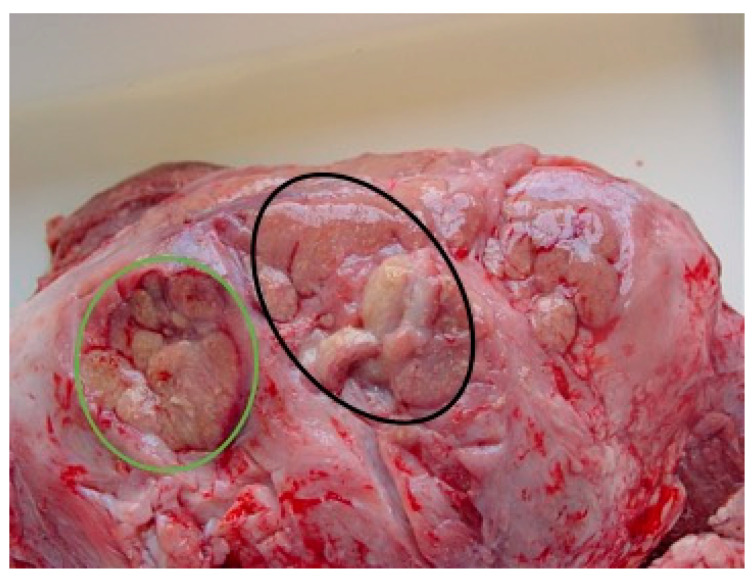
Granulomatous reaction in orbital cavity (green circle) and brain (black circle), respectively.

**Figure 3 vetsci-10-00311-f003:**
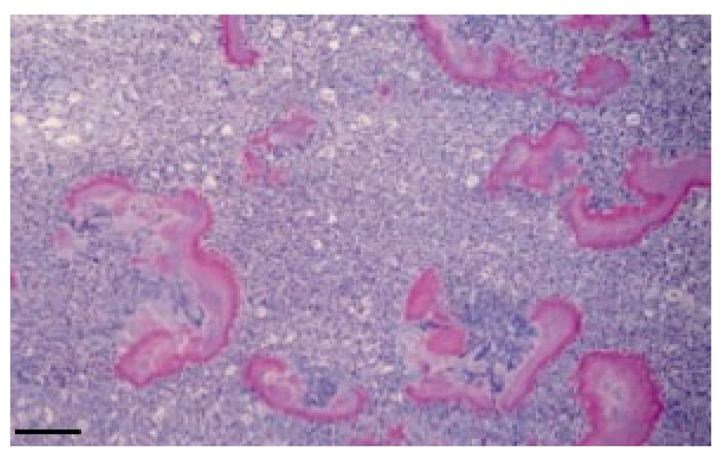
Typical histological features of granulomatous disease due to *A. lignieresii*. (brain; HE). Bar: 20 μm.

**Figure 4 vetsci-10-00311-f004:**
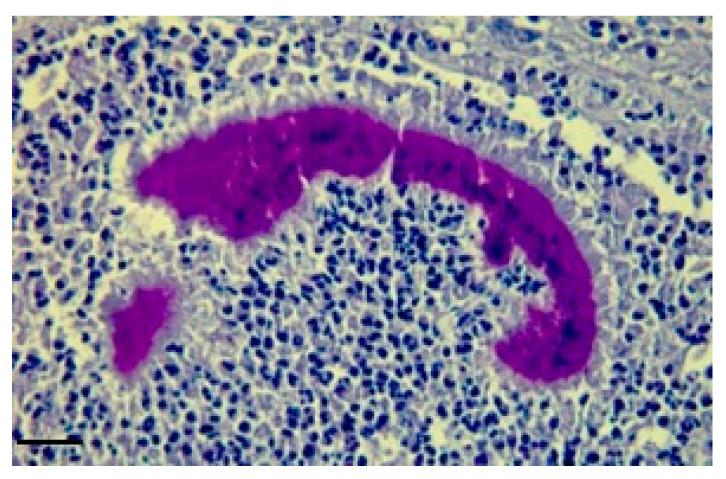
Granulomatous reaction results to be PAS-positive and the material arranged as a crown of clavate protrusions around each colony. (brain; PAS). Bar: 10 μm.

## Data Availability

Not applicable.

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
