# Peer review of "Infection Due to Actinobacillus lignieresii in Cattle with Brain and Ocular Involvement: Histological and Microbiological Features"

_vetsci, 2023, doi:10.3390/vetsci10050311_

Round 1
Reviewer 1 Report
This is an interesting case report about infection due to Actinobacillus lignieresii in cattle with brain and ocular involvement. The Authors focuses on histological and microbiological features of the lesions described.
The case reports of a well-known disease characterized by an unusual location of the lesions (eye and brain).
I provided all my comments and revisions in the text. I recommend the Authors to carefully revise the manuscript accordingly. They should improve the gross description to give proper evidence of the atypical location of the granulomatous lesions.
The conclusions should be rewritten. In the present form they are not in line with the subject of the manuscript.
English language needs to be carefully revised by a mother-tongue speaker. In some point English sentences are really incomprehensible.

Author Response
Replies to Reviewer 1
Dear Reviewer,
thanks for your the corrections. I made the requested changes and I had corrected the manuscript by a native English speaker. In particular:
I deleted the reference in the summary
I have inserted the citations and corrected the references
I added the breed
I described the lesion better
All changes have been made
Reviewer 2 Report
Major concerns:
Authors describe an interesting case of bovine cerebral and ocular actinobacillosis. Despite the case is well presented and the image are of good quality, the manuscript is not well written and most sentences have to be rewrite in a correct manner especially in the introduction and discussion sections. In addition, I suggest to check for a more updated references in the reference section.
Finally, for a correct diagnosis and identification I suggest to use bio-molecular assays (see Turni et al. 2019; https://doi.org/10.1111/avj.12868).
Have the authors tested the antibiotic susceptibility of the isolated strain? If not I suggest to perform it.
Minor concerns:
Line 20-22: Please remove this sentence. It is not possible to state.
Line 38: Please correct Lymphnodes with Lymph nodes.
Line 39-40: Please rephrase this sentence.
Line 43-44: Please remove this sentence. It is not possible to state.
Figure 3. Please choose a better image or adjust the focus of the image. The image dos not have the best focus.
Author Response
Dear Reviewer,
thanks for your the corrections. I made the requested changes and I had corrected the manuscript by a native English speaker.
Replies to Reviewer 2
Dear Reviewer,
thanks for your the corrections. I made the requested changes and I had corrected the manuscript by a native English speaker. In particular:
Line 20-22: Please remove this sentence. It is not possible to state. DONE
Line 38: Please correct Lymphnodes with Lymph nodes. DONE
Line 39-40: Please rephrase this sentence. DONE
Line 43-44: Please remove this sentence. It is not possible to state. DONE
Figure 3. Please choose a better image or adjust the focus of the image. The image dos not have the best focus.
We can't get a better focus of image.
6. For a correct diagnosis and identification I suggest to use bio-molecular assays (see Turni et al. 2019; https://doi.org/10.1111/avj.12868).
We have not carried out biomolecular assays, because the positive response to the bacteriological and histopathological examination has already allowed us to make a complete diagnosis
7. Have the authors tested the antibiotic susceptibility of the isolated strain? If not I suggest to perform it.
We tested the antibiotic susceptibility. Strain sensitive to cephalosporins, tetracyclines and fluoroquinolones. However, the encephalic localization casts doubt on the efficacy of any therapy
Reviewer 3 Report
The case report presented by the authors is interesting per se, but the manuscript has to be completely rewritten. The English style is full of mistakes ( e.g. line 57 granulomi is the plural form in Italian).The structure is not well designed, several parts of the case presentation should be moved to the discussion ( e.g. lines 98-102). Vice versa, the discussion contains several sentences that are more appropriate for the introduction.
Several terms are not properly used ( line 77: not colonies but aggregates).
The bibliography is not appropriate and updated. For istance , ( Line 156) why to mention for concurrent factors a paper published in 1902, regarding foot and mouth disease? At that time, the epidemiology and the knowledges about were completely different ( virus not discovered yet..)
. in my opinion, the manuscript cannot be accepted .
Author Response
Dear Reviewer,
thanks for your the corrections. I made the requested changes and I had corrected the manuscript by a native English speaker.
Reviewer 4 Report
A brief summary
The authors report a case of bovine actinobacillosis with ocular and cerebellar lesions. The cow had presented neurological signs and was suspected to have BSE. The diagnosis was confirmed with thorough histopathological and bacteriological examination. This is the first scientific report of Actinobacillus lignieresii infection with such a localization. The of the finding is discussed with existing literature.
General comments
The finding is thoroughly documented, methodology is well described, and the histopathology is presented with high quality photographs. The manuscript requires improvement. Bacteriological methods lack molecular approach in discussion, and, in general, the role of bacteriology should be more pronounced as the topic is A. lignieresii infection. The sentence in lines 22-25 and 110-113 is too self-evident to be mentioned in a scientific publication. Introduction should better focus to the disease in cattle, prevalence and importance from welfare and economic point of view. Discussion contains relevant literature, but it could be written more concisely and from the point of view of the reported case. There is some repetition in different paragraphs, the citations from different articles could be merged better. Zoonosis aspect, which is most theoretical in this case, should be discussed less. The last paragraph of discussion deals too much with general zoonotic and one health issues (lines 235-245) and could be omitted. The abstract is not coherent with discussion with regards to zoonosis aspect. In general, a better organization and focusing of the text would improve the manuscript.
Specific comments
There are unclear sentences: lines 79-80 The surrounded by…; 113-114 The histological… ;
Author Response

(The authors gave the same response as above.)

Round 2
Reviewer 1 Report
The Authors have properly revised the manuscript as suggested. However, the text between lines 331 to 341 (conclusions) should be deleted because it is not coherent with the topic of the manuscript. It is a general statement on zoonosis and One Health not related to the manuscript. Once removed, the article can be accepted.
Author Response
Response to reviewer 1:
I have deleted lines in the conclusions as requested.
Reviewer 2 Report
I suggest to remove this kind of sentence: "this is the first case" throughout the paper. It is not possible to state.
Author Response
Response to reviewer 2:
I have remove this kind of sentence: "this is the first case" throughout the paper
Reviewer 4 Report
Comment to the revised version
Short summary and abstract must be completely revised. They must briefly summarize the findings of the study, and be independent from each other. The version 1 abstract was much better than the revised one.
The conclusion must be completely revised. It should briefly outline the take-home message from your study, ie. the first paragraph in the text. Remove the paragraph about one health as the zoonosis aspect is discussed in the discussion. Treatment and vaccination do not belong to the conclusion because they have not been studied. These paragraphs should also be removed.
Specific comments
lines 105-108 are difficult to understand, require revision.
Author Response
Dear reviewer,
I made the requested changes.
The short summary and abstract had already been completely reviewed during the first review phase. Therefore, bringing them back to version 1 would go against what the other reviewers requested.
Response to reviewer 4:
1. The short summary and abstract had already been completely reviewed during the first review phase. Therefore, bringing them back to version 1 would go against what the other reviewers requested.
2. The conclusion was re vised as also requested by reviewer 1.
3. I revised lines 105-108.